# IFN-λ prevents influenza virus spread from the upper airways to the lungs and limits virus transmission

Jonas Klinkhammer[1,2†], Daniel Schnepf[1,3†], Liang Ye[1], Marilena Schwaderlapp[1], Hans Henrik Gad[4], Rune Hartmann[4], Dominique Garcin[5], Tanel Mahlakõiv[1‡], Peter Staeheli[1,6*]

[1]Institute of Virology, Medical Center, University of Freiburg, Freiburg, Germany; [2]MOTI-VATE Graduate School, Medical Center, University of Freiburg, Freiburg, Germany; [3]Spemann Graduate School of Biology and Medicine (SGBM), Albert Ludwigs University Freiburg, Freiburg, Germany; [4]Department of Molecular Biology and Genetics, Aarhus University, Aarhus, Denmark; [5]Department of Microbiology and Molecular Medicine, University of Geneva, Geneva, Switzerland; [6]Faculty of Medicine, University of Freiburg, Freiburg, Germany

**\*For correspondence:**
peter.staeheli@uniklinik-freiburg.de

[†]These authors contributed equally to this work

**Present address:** [‡]Department of Microbiology and Immunology, Weill Cornell Medicine, New York, United States

**Competing interests:** The authors declare that no competing interests exist.

**Abstract** Host factors restricting the transmission of respiratory viruses are poorly characterized. We analyzed the contribution of type I and type III interferon (IFN) using a mouse model in which the virus is selectively administered to the upper airways, mimicking a natural respiratory virus infection. Mice lacking functional IFN-λ receptors ($Ifnlr1^{-/-}$) no longer restricted virus dissemination from the upper airways to the lungs. $Ifnlr1^{-/-}$ mice shed significantly more infectious virus particles via the nostrils and transmitted the virus much more efficiently to naïve contacts compared with wild-type mice or mice lacking functional type I IFN receptors. Prophylactic treatment with IFN-α or IFN-λ inhibited initial virus replication in all parts of the respiratory tract, but only IFN-λ conferred long-lasting antiviral protection in the upper airways and blocked virus transmission. Thus, IFN-λ has a decisive and non-redundant function in the upper airways that greatly limits transmission of respiratory viruses to naïve contacts.
DOI: https://doi.org/10.7554/eLife.33354.001

## Introduction

Influenza and other respiratory viruses are readily transmitted in community settings. The resulting infection chains and concomitant diseases represent an enormous economic and public health burden (*Osterhaus et al., 2015*). At present, it is largely unknown if and how the innate immune response influences virus transmission efficacy. However, a better understanding of this issue is likely to lead to new therapeutic strategies that reduce viral transmission from infected individuals to naïve contacts.

Type I and type III interferon (IFN) are virus-induced cytokines that potently restrict viral replication during the first days of infection before activation of the adaptive immune system occurs (*Lazear et al., 2015*; *Wack et al., 2015*). The type I IFN family consists of several IFN-α subtypes, a single IFN-β and several minor family members that all bind to and act via the IFN-α/β receptor complex (IFNAR), which is expressed on most nucleated cells (*Lazear et al., 2015*; *Wack et al., 2015*) with the possible exception of intestinal epithelial cells (*Lin et al., 2016*; *Mahlakõiv et al., 2015*). The members of the type III IFN family (IFN-λ) bind to a different receptor complex (IFN-λ receptor; IFNLR), which is highly expressed on epithelial cells (*Kotenko et al., 2003*; *Lazear et al., 2015*;

**eLife digest** Influenza ('the flu') and other respiratory viruses make millions of people ill every year, placing a large burden on the healthcare system and the economy. Unfortunately, few options for preventing or treating these infections currently exist.

The flu virus spreads from infected individuals, enters a new host through the nose and establishes an infection in the upper airways. If the infection stays restricted to this region of the respiratory tract – which consists of the nasal cavity, sinuses, throat and larynx – it causes a rather mild disease. However, if it spreads to the lungs it can cause potentially life-threatening viral pneumonia.

Epithelial cells line the upper respiratory tract, forming a physical border between the outside world and the human body. These cells are therefore the first to face the incoming virus. In response, the epithelial cells release messenger molecules termed interferons that warn nearby cells to increase their antiviral defenses.

There are several subtypes of interferons, such as IFN-α, IFN-β and IFN-λ, but it was not known how each subtype helps to combat respiratory viruses. To investigate, Klinkhammer, Schnepf et al. exposed mice to flu viruses in a way that mimicked how an infection would naturally start in the upper airways in humans. Some of the mice were genetically engineered so that they could not respond to either IFN-α/β or IFN-λ.

The virus spread most effectively from the nasal cavity to the lungs in mice whose IFN-λ system was defective. Infections in mice that lacked IFN-λ were also more likely to spread to other individuals. Furthermore, treating mice with IFN-λ, but not IFN-α, gave their upper respiratory tract long-lasting protection against flu infections and prevented the spread of the virus.

IFN-λ therefore has a specific and significant role in protecting the upper airways against viruses, and could potentially be used as a drug to block the spread of infections between humans. Currently, IFN-λ is in clinical trials as a potential treatment for hepatitis D. To repurpose it for upper respiratory tract infections, its effectiveness against specific respiratory viruses will first have to be evaluated.

DOI: https://doi.org/10.7554/eLife.33354.002

*Sheppard et al., 2003*; *Sommereyns et al., 2008*; *Wack et al., 2015*) and neutrophils (*Blazek et al., 2015*; *Broggi et al., 2017*; *Espinosa et al., 2017*; *Galani et al., 2017*).

Transmission of influenza and other respiratory viruses is traditionally studied in ferrets and occasionally in guinea pigs (*Bouvier, 2015*). In contrast to mice, these animals are not readily accessible to genetic manipulation, and the role of IFN and IFN-regulated host restriction factors is not easily dissected. A recent report provided evidence that the mouse represents a good alternative animal model for studying influenza virus transmission, as long as suitable virus strains are employed (*Edenborough et al., 2012*; *Ivinson et al., 2017*).

Influenza virus replication is strongly restricted by IFN-regulated *Mx* genes (*Haller et al., 2015*). Since most standard inbred mouse strains carry defective *Mx* alleles (*Haller et al., 2015*), the full magnitude of influenza virus restriction by the IFN system only becomes apparent when mouse strains carrying functional *Mx1* alleles derived from wild mice are employed (*Haller et al., 2015*). We previously introduced defective alleles of the type I IFN receptor α chain ($Ifnar1^{-/-}$) or the IFN-λ receptor one chain ($Ifnlr1^{-/-}$) into *Mx1*-competent C57BL/6 mice and thereby generated a set of mouse lines that carry functional *Mx1* alleles but differ in their ability to respond to type I IFN and IFN-λ (*Mordstein et al., 2008*). Using these knockout mouse strains we demonstrated that type I IFN plays a prominent role in the defense against respiratory viruses, including influenza viruses, respiratory syncytial virus or human metapneumovirus, whereas protection mediated by IFN-λ was surprisingly small (*Mordstein et al., 2010b*). In contrast, the protective role of IFN-λ during viral infections of the small intestine was found to be far more pronounced than that of type I IFN (*Baldridge et al., 2017*; *Baldridge et al., 2015*; *Mahlakõiv et al., 2015*; *Nice et al., 2015*; *Pott et al., 2011*). This suggested either that (i) the gut and the respiratory tract use fundamentally different antiviral defense strategies or (ii) IFN-λ is important in both organ systems but its beneficial role in the respiratory tract cannot be demonstrated easily due to experimental limitations.

Recent work with influenza virus-infected mice (*Galani et al., 2017*) indicated that IFN-λ is produced more quickly than type I IFN, suggesting that IFN-λ plays a non-redundant role in suppressing early virus growth in the respiratory tract. Nevertheless, protective effects of IFN-λ were observed only if very low infection doses were used, presumably because mice with defective *Mx1* alleles were employed for these studies. Furthermore, the important question whether IFN-λ might play a role in restricting transmission of respiratory viruses from infected animals to naïve contacts was not addressed.

According to standard protocols for experimental influenza virus infections, the virus is delivered intranasally to anesthetized mice in a 40–50 µl volume, allowing immediate infection of the entire respiratory tract. This experimental setup might override the natural barrier function of the nasal mucosa by allowing the infection to artificially initiate in the lower respiratory tract (*Ivinson et al., 2017*). In this study, we applied an infection protocol that guarantees a selective delivery of the virus inoculum to the upper respiratory tract, thus more closely mimicking the natural course of infection. Under such experimental conditions, virus transmission to naïve contacts as well as virus spread from the upper airways to the lower respiratory tract was inhibited far more efficiently by IFN-λ than by type I IFN. Interestingly, the antiviral effect induced by type I IFN in the upper airways was more transient than that of IFN-λ, while no such difference was observed in the lungs. Furthermore, a small number of epithelial cells in the proximal upper respiratory tract failed to respond sufficiently to type I IFN but did respond to IFN-λ. These observations provide a mechanistic explanation for the superior role of IFN-λ in the upper airways.

## Results

### IFN-λ prevents influenza virus spread from the upper respiratory tract to the lungs

Most previous studies failed to assign a prominent role to IFN-λ in the resistance of mice against influenza and other respiratory viruses (*Galani et al., 2017*; *Mordstein et al., 2008*). Similarly, when we applied $10^4$ PFU of the H7N7 influenza A virus strain SC35M intranasally in a standard 40 µl volume, we noted that *Mx1*-competent animals carrying a defective IFN-λ receptor (*Ifnlr1$^{-/-}$*) supported similar virus replication in the lungs on day three post infection, compared with *Mx1*-competent wild-type (WT) animals. In contrast, *Mx1*-competent mice lacking functional type I IFN receptors (*Ifnar1$^{-/-}$*) showed significantly elevated viral titers under such experimental conditions (*Figure 1A*). Respiratory viruses enter the human body via the upper respiratory tract, and the majority of acute respiratory viral infections in humans are confined to the upper airways (*Cotton et al., 2008*). To mimic a natural infection in which virus replication initiates in the upper airways, we established an upper respiratory tract infection model in mice. Using poliovirus that is unable to infect mouse cells as an indicator we found that virus delivery can be targeted specifically to the upper respiratory tract of mice if the inoculum is applied in a volume of 10 µl (*Figure 1—figure supplement 1*). In contrast, a large proportion of the inoculum reached the lungs if the inoculum was applied in a volume of 40 µl. When a 10 µl volume was used to deliver influenza virus strain SC35M specifically to the upper respiratory tract of WT mice, the virus grew well in the upper airways, but was absent or present at only very low titers in the tracheae (3 of 22 animals) and the lungs (1 of 22 animals) on day five post-infection (*Figure 1B*). In *Ifnlr1$^{-/-}$* mice, viral titers in the upper airways were significantly higher than in WT mice and, most interestingly, many infected *Ifnlr1$^{-/-}$* mice contained high virus levels in the tracheae (11/23) and lungs (13/23) (*Figure 1B*). As expected, *Ifnar1$^{-/-}$* mice also had significantly higher virus titers in the upper airways than WT mice, and more animals contained virus in the tracheae (5/23) and lungs (9/23) (*Figure 1B*). In comparison, *Ifnar1$^{-/-}$* mice seemed to control the virus spread from the upper airways to the lungs slightly better than *Ifnlr1$^{-/-}$* mice (*Figure 1B*), indicating that IFN-λ is more potent in containing viral infections within the upper respiratory tract than type I IFN.

To determine whether this unexpected phenotype of *Ifnlr1$^{-/-}$* mice is restricted to the SC35M virus, we performed similar infection experiments with the H3N2 influenza A virus strain Udorn. After selective delivery to the upper respiratory tract, Udorn virus was rarely found in the tracheae (1/16) or lungs (0/16) of WT mice, but was frequently present in the tracheae (15/16) and lungs (14/16) of *Ifnlr1$^{-/-}$* mice (*Figure 1C*). *Ifnar1$^{-/-}$* mice showed an intermediate phenotype under these

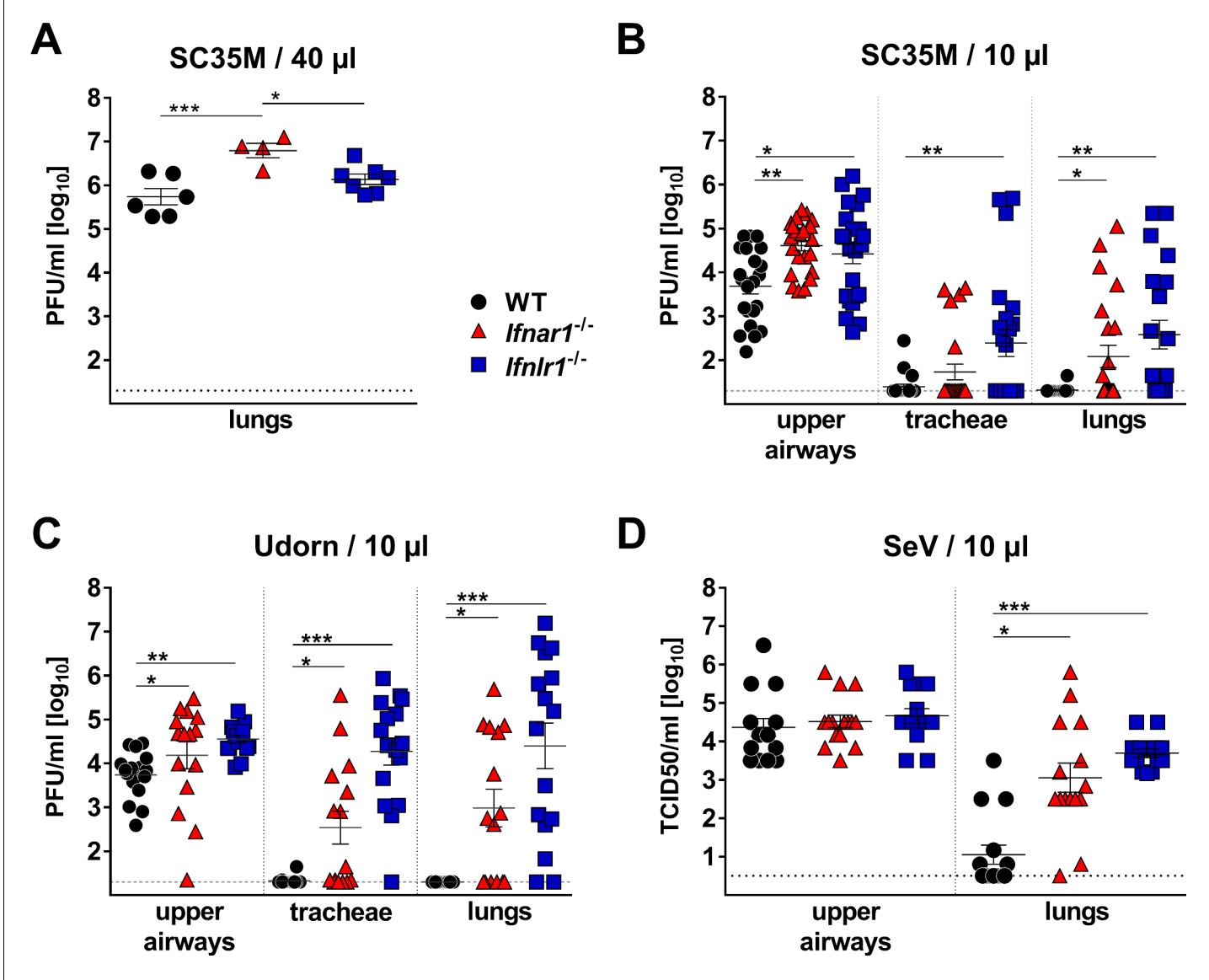

**Figure 1.** IFN-λ prevents virus spread from the upper airways to the lungs of mice. (**A**) Standard intranasal delivery of virus inoculum: WT (n = 6), *Ifnar1*$^{-/-}$ (n = 4) and *Ifnlr1*$^{-/-}$ (n = 7) mice were intranasally infected with $10^4$ PFU of SC35M in a volume of 40 µl, and viral titers in the lungs were determined on day three post infection by plaque assay. (**B–D**) Selective virus delivery to the upper respiratory tract: (**B**) WT (n = 22), *Ifnar1*$^{-/-}$ (n = 23) and *Ifnlr1*$^{-/-}$ (n = 23) mice were intranasally infected with $10^4$ PFU of SC35M in a volume of 10 µl. Mice were sacrificed on day five post infection, and viral titers in the upper airways, tracheae and lungs were determined by plaque assay. Pooled results from three independent experiments are shown. (**C**) WT (n = 16), *Ifnar1*$^{-/-}$ (n = 15) and *Ifnlr1*$^{-/-}$ (n = 16) mice were intranasally infected with $5 \times 10^3$ PFU of Udorn in a volume of 10 µl. Mice were sacrificed on day five post infection, and viral titers in the upper airways, tracheae and lungs were determined by plaque assay. Pooled results from two independent experiments are shown. (**D**) WT (n = 15), *Ifnar1*$^{-/-}$ (n = 15) and *Ifnlr1*$^{-/-}$ (n = 15) mice were intranasally infected with $10^3$ TCID$_{50}$ of SeV in a volume of 10 µl. Mice were sacrificed on day five post infection, and viral titers in the upper airways and lungs were determined by the TCID$_{50}$ method. Pooled results from two independent experiments are shown. Symbols represent individual mice, and bars represent means ± SEM. Statistical analysis: One-way ANOVA with Tukey's multiple comparisons was used to compare viral titers in the upper airways: asterisks indicate p-values: ***p<0.001, **p<0.01, *p<0.05. Fisher's exact test was used to compare events of virus spread: circles indicate p-values: °°°p<0.001, °°p<0.01, °p<0.05.

DOI: https://doi.org/10.7554/eLife.33354.003

The following figure supplement is available for figure 1:

**Figure supplement 1.** Selective infection of the upper respiratory tract can be achieved by applying the virus inoculum in a small volume.

DOI: https://doi.org/10.7554/eLife.33354.004

experimental conditions, and infectious virus was found at lower frequency in the tracheae (8/15) and lungs (9/15) of *Ifnar1*$^{-/-}$ mice compared with *Ifnlr1*$^{-/-}$ mice. A similar picture emerged when a murine respiratory virus (Sendai virus; SeV) was employed in infection experiments. SeV reached the lungs of all (15/15) *Ifnlr1*$^{-/-}$ mice when selectively applied to the upper respiratory tract (*Figure 1D*). In contrast, SeV reached the lungs of WT mice at significantly reduced frequency (6/15) under these experimental conditions, and virus titers at day five post-infection in the lungs of WT mice were generally lower compared with *Ifnlr1*$^{-/-}$ mice. Taken together, these data clearly indicated that IFN-λ is an indispensable antiviral cytokine that restricts respiratory viral infections to the upper airways and limits virus spread to the lungs.

To better characterize the early events of the antiviral defense in the upper airways, we measured the baseline expression levels of various IFN genes. Expression of the IFN-λ2 and IFN-λ3 genes was significantly reduced in snout tissue of *Ifnar1*$^{-/-}$ mice compared with WT and *Ifnlr1*$^{-/-}$ mice, whereas baseline expression of IFN-α and IFN-β genes was comparable (*Figure 2A*). Consequently, baseline expression of the *Mx1* gene was also lower in snout tissue of *Ifnar1*$^{-/-}$ mice than in WT and *Ifnlr1*$^{-/-}$ mice (*Figure 2A*). Upon infection with Udorn virus, expression of the IFN-λ genes in *Ifnar1*$^{-/-}$ mice returned to comparable levels of WT and *Ifnlr1*$^{-/-}$ mice within two days (*Figure 2B*). Interestingly, expression of the IFN-λ genes in snout homogenates of WT and *Ifnlr1*$^{-/-}$ mice was only slightly induced after virus infection, whereas expression of the IFN-β gene was induced about tenfold in all mouse strains (*Figure 2B*). These data indicate that functional type I IFN signaling is needed for

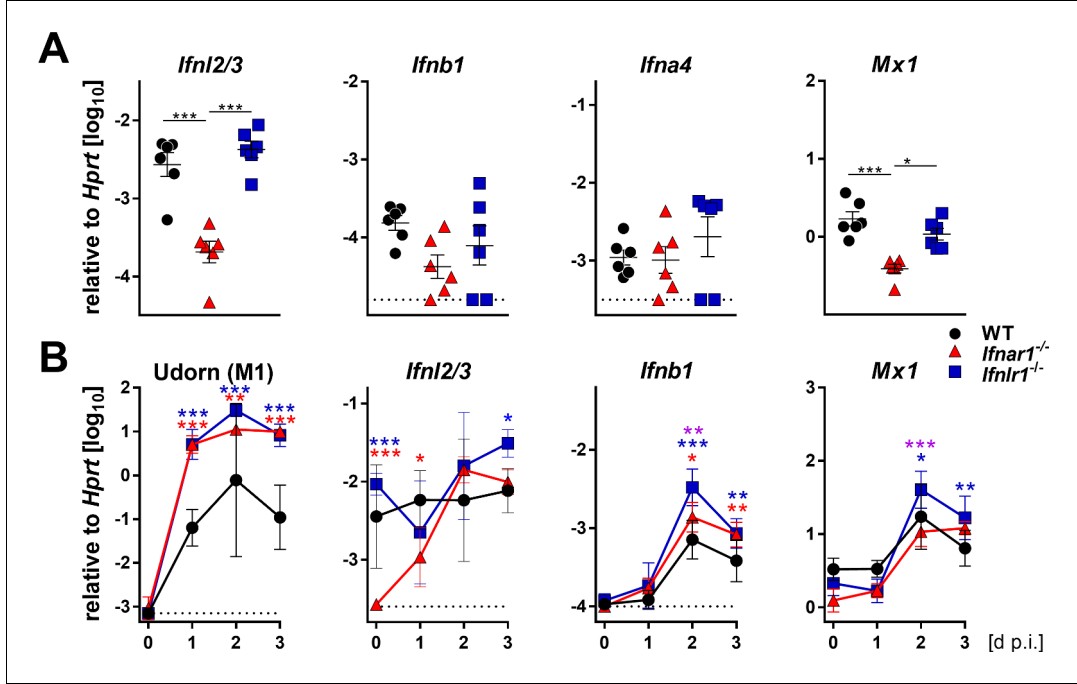

**Figure 2.** Basal expression of IFN-λ genes is reduced in *Ifnar1*$^{-/-}$ mice. (**A**) Basal expression of type I (*Ifnb1* and *Ifna4*), type III IFNs (*Ifnl2/3*) and *Mx1* was measured by RT-qPCR in snout homogenates of WT (n = 6), *Ifnar1*$^{-/-}$ (n = 6) and *Ifnlr1*$^{-/-}$ (n = 6). Gene expression levels are shown relative to the housekeeping gene *Hprt*. Symbols represent individual mice, and bars represent means ± SEM. Statistical analysis: One-way ANOVA with Tukey's multiple comparisons; asterisks indicate p-values: ***p<0.001, *p<0.05. (**B**) WT (n = 21), *Ifnar1*$^{-/-}$ (n = 22) and *Ifnlr1*$^{-/-}$ (n = 23) mice were intranasally infected with $10^4$ PFU of Udorn in a volume of 10 µl. Mice were sacrificed at the indicated time points (n = 5–7) and snout homogenates were processed for RT-qPCR. Expression levels of mRNAs encoding viral M1 protein or cellular gene products IFN-λ2/3, IFN-β and Mx1 are shown relative to transcription of the *Hprt* housekeeping gene. Symbols represent means ± SD. Red or blue asterisks indicate statistically significant differences between WT and *Ifnar1*$^{-/-}$ or *Ifnlr1*$^{-/-}$, respectively; purple asterisks indicate differences between *Ifnar1*$^{-/-}$ and *Ifnlr1*$^{-/-}$. Statistical analysis: Two-way ANOVA; asterisks indicate p-values: ***p<0.001, **p<0.01, *p<0.05.

DOI: https://doi.org/10.7554/eLife.33354.005

proper baseline expression of the IFN-λ genes, which contributes to early virus defense in the upper airways.

When the infection was initiated by applying the inoculum selectively to the upper airways, none of the influenza viruses used in this study caused morbidity or mortality, although virus titers in lungs of $Ifnlr1^{-/-}$ mice reached levels above $10^6$ PFU in some animals on day five post-infection (*Figure 1B*). These observations are in good agreement with recent work indicating that upper airway infections with influenza viruses are usually benign in mice presumably because saliva components delay the spread of influenza viruses to the lungs (*Ivinson et al., 2017*). Delayed virus arrival in the lungs may permit timely adaptive immune responses to eliminate the virus before it causes irreversible tissue damage. We concluded from these observations that the upper airway infection model is not suitable for studying influenza virus-induced pathology in mice, but it allows addressing the question which factors might control virus growth at the entry site.

## High virus loads in nasal excretions of infected mice lacking functional IFN-λ receptors

The efficacy of respiratory virus transmission depends, among other parameters, on the production of mucosal secretions or aerosols containing a sufficiently high number of infectious particles (*Herfst et al., 2017*). To determine whether nasal excretions of $Ifnlr1^{-/-}$ mice contain more infectious virus than excretions of WT mice, we sampled the nostrils of Udorn-infected mice with wet cotton swabs. This analysis revealed that $Ifnlr1^{-/-}$ mice shed significantly more infectious virus than WT or $Ifnar1^{-/-}$ mice (*Figure 3A*). Already at 12 hr post infection, $Ifnlr1^{-/-}$ mice secreted remarkable amounts of infectious virus. Interestingly, virus shedding was only transient and peaked between 24 and 36 hr post infection (*Figure 3A*). When the nostrils of SeV-infected mice were swabbed (*Figure 3B*), we found at least 10-fold more infectious SeV in mucosal secretions of $Ifnlr1^{-/-}$ mice compared with WT controls between days 2 and 4 post infection. Levels of infectious SeV in mucosal secretions of $Ifnar1^{-/-}$ mice were only slightly increased compared with WT mice. Importantly, $Ifnar1^{-/-}$ mice shed significantly less virus than $Ifnlr1^{-/-}$ mice at all time points between days 2 and 4 post infection (*Figure 3B*), indicating that IFN-λ is the dominant limiting factor for virus shedding.

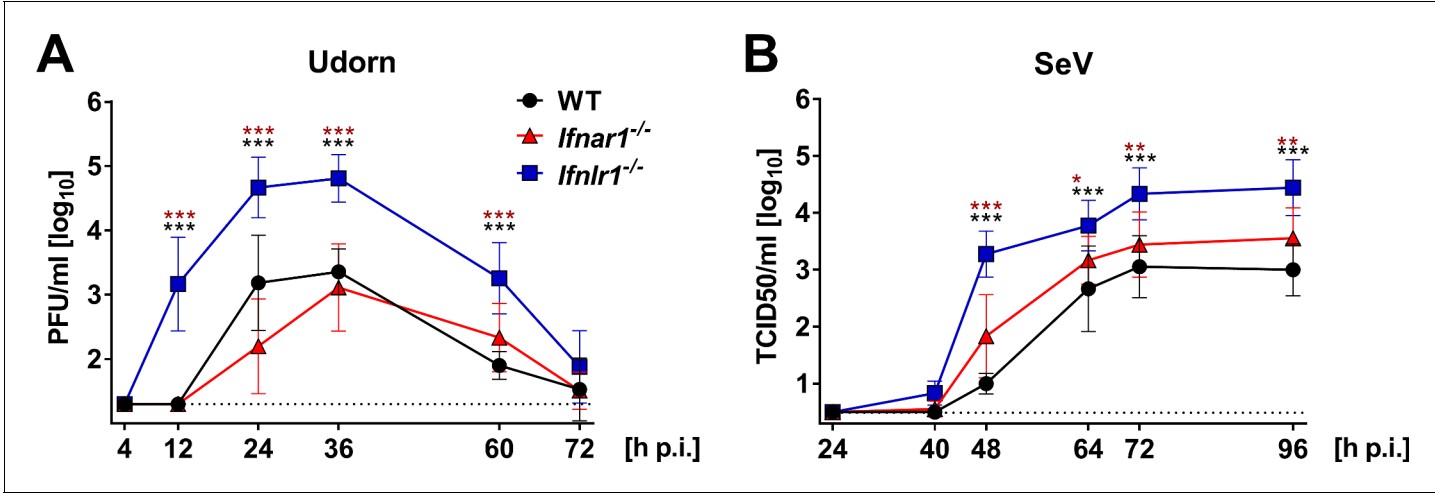

**Figure 3.** $Ifnlr1^{-/-}$ mice secrete high amounts of infectious virus. (**A**) WT (n = 7), $Ifnar1^{-/-}$ (n = 9) and $Ifnlr1^{-/-}$ (n = 9) mice were intranasally infected with $10^5$ PFU of Udorn in a volume of 10 µl. (**B**) WT (n = 6), $Ifnar1^{-/-}$ (n = 6) and $Ifnlr1^{-/-}$ (n = 6) mice were intranasally infected with $10^3$ TCID$_{50}$ of SeV in a 10 µl volume. Nasal swabs were taken at the indicated time points post infection. Infectious virus recovered from the swabs was quantified by plaque assay (Udorn) or the TCID$_{50}$ method (SeV). Symbols represent means ±SD. Statistical analysis: Two-way ANOVA; black asterisks indicate significant differences between WT and $Ifnlr1^{-/-}$, red asterisks indicate significant differences between $Ifnar1^{-/-}$ and $Ifnlr1^{-/-}$. P-values: ***p<0.001, **p<0.01, *p<0.05.
DOI: https://doi.org/10.7554/eLife.33354.006

## IFN-λ receptor-deficient mice readily transmit respiratory viruses to contact animals

Next we evaluated whether increased virus shedding by $Ifnlr1^{-/-}$ mice would translate into enhanced spread of respiratory viruses among mice. We set up transmission experiments in which infected mice were cohoused with highly susceptible sentinel mice that lack functional receptors for type I IFN and IFN-λ ($Ifnar1^{-/-}$ $Ifnlr1^{-/-}$). In a first series of experiments, we infected WT, $Ifnar1^{-/-}$ or $Ifnlr1^{-/-}$ mice with $10^5$ PFU of Udorn. To mimic natural virus transmission in a family setting, groups of three infected mice were cohoused with four sentinels in a single cage for a period of four days. Viral titers were then determined in the upper airways of the sentinels. Under such experimental conditions, 79% of the sentinels that were in contact with infected $Ifnlr1^{-/-}$ mice became infected (*Figure 4A*). As expected from the virus excretion data (*Figure 3A*), transmission of Udorn to sentinel mice was observed much less frequently (17% and 42%, respectively) when infected WT or $Ifnar1^{-/-}$ mice were employed as virus spreaders (*Figure 4A*).

In a second experiment we infected WT, $Ifnar1^{-/-}$ or $Ifnlr1^{-/-}$ mice with $10^5$ PFU of a different H3N2 influenza A virus strain (HK68) before cohousing with $Ifnar1^{-/-}$ $Ifnlr1^{-/-}$ double-deficient sentinels. Under these experimental conditions, 10 of 11 (91%) sentinels in contact with infected $Ifnlr1^{-/-}$ mice contracted the virus, whereas only 27% of the sentinels cohoused either with infected WT or $Ifnar1^{-/-}$ mice became infected (*Figure 4B*). To determine whether transmission of other respiratory viruses might also be restricted by IFN-λ, we studied mouse-to-mouse transmission of SeV using similar experimental conditions, except that we cohoused each infected mouse individually with one $Ifnar1^{-/-}$ $Ifnlr1^{-/-}$ sentinel animal. All sentinels co-housed with SeV-infected $Ifnlr1^{-/-}$ mice contracted the virus, whereas only 38% of the sentinels cohoused with WT mice and 75% of the sentinels cohoused with $Ifnar1^{-/-}$ became infected with SeV (*Figure 4C*). Together, these data show that IFN-λ effectively limits the spread of respiratory viruses in a contact transmission setting.

## IFN-λ is the main mediator of antiviral defense in the upper respiratory tract

Transmission of respiratory viruses between humans presumably occurs at low doses. A recent study in mice suggests that endogenous IFN-λ confers better protection against influenza virus infection compared with type I IFN if very low viral doses are used (*Galani et al., 2017*). In our experiments, viral titer differences in the upper airways of WT and $Ifnlr1^{-/-}$ mice were relatively small (*Figure 1*). This suggested that the high infection dose used in those experiments had largely masked the protective effect of IFN-λ. Indeed, when the challenge dose of Udorn was reduced to 100 PFU per animal, the differences in the upper airways of WT and $Ifnlr1^{-/-}$ mice became more obvious

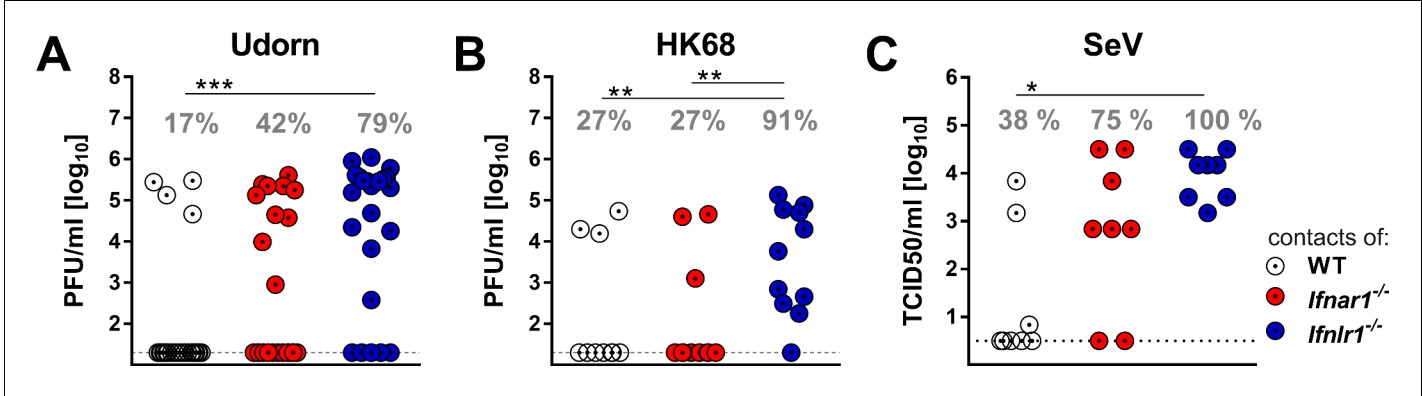

**Figure 4.** IFN-λ limits contact transmission of respiratory viruses among mice. WT, $Ifnar1^{-/-}$ and $Ifnlr1^{-/-}$ mice were intranasally infected with (A) $10^5$ PFU of Udorn, (B) $10^5$ PFU of HK68 or (C) $10^3$ TCID$_{50}$ of SeV in a volume of 10 µl. At 24 hr post infection, the infected mice were cohoused with naïve $Ifnar1^{-/-}Ifnlr1^{-/-}$ contact mice for 4 days (Udorn and HK68) or 6 days (SeV). Viral titers in the upper airways of individual contact mice are plotted, and calculated rates of successful virus transmission are indicated. Statistical analysis: Fisher's exact test; asterisks indicate p-values: ***p<0.001, **p<0.01, *p<0.05.

DOI: https://doi.org/10.7554/eLife.33354.007

(*Figure 5—figure supplement 1*). In agreement with the virus excretion data (*Figure 3*), which had indicated a minor role for type I IFN in this process, replication of Udorn in the upper airways was not increased in *Ifnar1*$^{-/-}$ animals when compared with WT mice (*Figure 5—figure supplement 1*), supporting the view that IFN-λ plays a more prominent role in antiviral protection of the upper respiratory tract than type I IFN.

To confirm that IFN-λ is more potent than type I IFN in the upper respiratory tract, we used a reciprocal experimental approach. We compared the antiviral potency of a broadly cross-reactive hybrid IFN-αB/D and mouse IFN-λ2 prophylactically applied into the airways. Pilot experiments with differentiated primary mouse airway epithelial cell cultures, known to express functional receptors for both type I and type III IFN (*Crotta et al., 2013*), demonstrated that hybrid IFN-αB/D and mouse IFN-λ2 can induce the IFN-responsive genes *Isg15, Stat1* and *Mx1* to similar levels if used at identical concentrations (*Figure 5—figure supplement 2*). To exclude unwanted interference from virus-induced endogenous IFN, we used *Ifnlr1*$^{-/-}$ and *Ifnar1*$^{-/-}$ mice and treated them intranasally with IFN-α or IFN-λ, respectively, one day before infection with the Udorn virus. The infection was performed with a 40 µl volume to ensure virus delivery to all parts of the respiratory tract. IFN-λ pretreatment potently inhibited virus replication in the upper airways, whereas IFN-α pretreatment had no such effect (*Figure 5A*). Of note, the differences between IFN-α and IFN-λ were far less pronounced in the lungs (*Figure 5A*).

Next, we determined whether the replication of SeV in the upper respiratory tract of mice is affected by IFN-λ and type I IFN in a similar manner as observed for influenza virus. WT mice were treated intranasally with identical doses of IFN-α or IFN-λ one day before infection with SeV and co-housed with *Ifnar1*$^{-/-}$ *Ifnlr1*$^{-/-}$ double-deficient sentinel mice. On day six post infection with SeV, viral titers in the upper airway of IFN-λ-treated mice were on average about 5-fold lower than in mock- or IFN-α-treated mice (*Figure 5B*). Only 33% (3/9) of the IFN-λ-treated mice transmitted SeV to sentinel animals, whereas 67% (6/9) of the IFN-α-treated mice infected the sentinels upon co-housing (*Figure 5C*). Mock-treated WT mice transmitted SeV at a frequency of 88% (7/8) under the same conditions (*Figure 5C*). Taken together, these experiments demonstrated that IFN-λ is able to inhibit virus replication in the upper respiratory tract to a much greater extent than IFN-α. Furthermore, only IFN-λ strongly decreased the rate of successful virus transmission to cage mates.

## IFN-λ confers long-lasting virus protection in the upper airways, whereas IFN-α-mediated protection is short-lived

To investigate the possibility that IFN-α might not efficiently reach the epithelial cells of the upper airways, we applied the various IFN preparations via the subcutaneous route before the mice were infected with SeV in a 40 µl volume to ensure infection of the entire respiratory tract. IFN-λ applied subcutaneously provided strong and long-lasting protection against SeV in the upper airways and lungs of mice (*Figure 6A*). This result is consistent with experiments in which IFN-λ was applied intranasally (*Figure 5B*). In contrast, IFN-α-treated animals sacrificed on either day 4 or six post infection had similar levels of SeV in nasal swabs and upper airways as mock-treated mice, but the IFN-α-treated animals contained reduced viral titers in the lungs (*Figure 6A*). Interestingly, IFN-α-treated animals sacrificed on day two post-infection showed significantly reduced viral titers in swabs and upper airways compared with mock-treated controls (*Figure 6A*). Thus, although this latter observation demonstrated that IFN-α is anti-virally active in the upper airways irrespective of the application route, our data clearly indicated that the antiviral effect of IFN-α, specifically in the upper airways, is surprisingly short-lived.

In a second experiment we strove to extend these findings and asked whether IFN-α might similarly inhibit the replication of influenza virus in the upper airways only transiently. In this experiment, the IFN preparations were applied by the intranasal route. Indeed, IFN-α and IFN-λ both exhibited pronounced inhibitory effects on Udorn virus replication in the upper airways at 24 hr post infection (*Figure 6B*, left panel). However, the antiviral effect of IFN-α in the upper airways could no longer be detected at 72 hr post infection, while virus inhibition by IFN-λ remained prominent (*Figure 6B*, right panel). Interestingly, there were no differences between the antiviral effects of IFN-α and IFN-λ in the lungs at 72 hr post infection (*Figure 6B*), indicating that the short-lived nature of the type I IFN response represents a unique feature of the upper respiratory tract. In concordance with these data, we found that IFN-α applied subcutaneously at doses ranging from 0.1 to 1.0 µg failed to

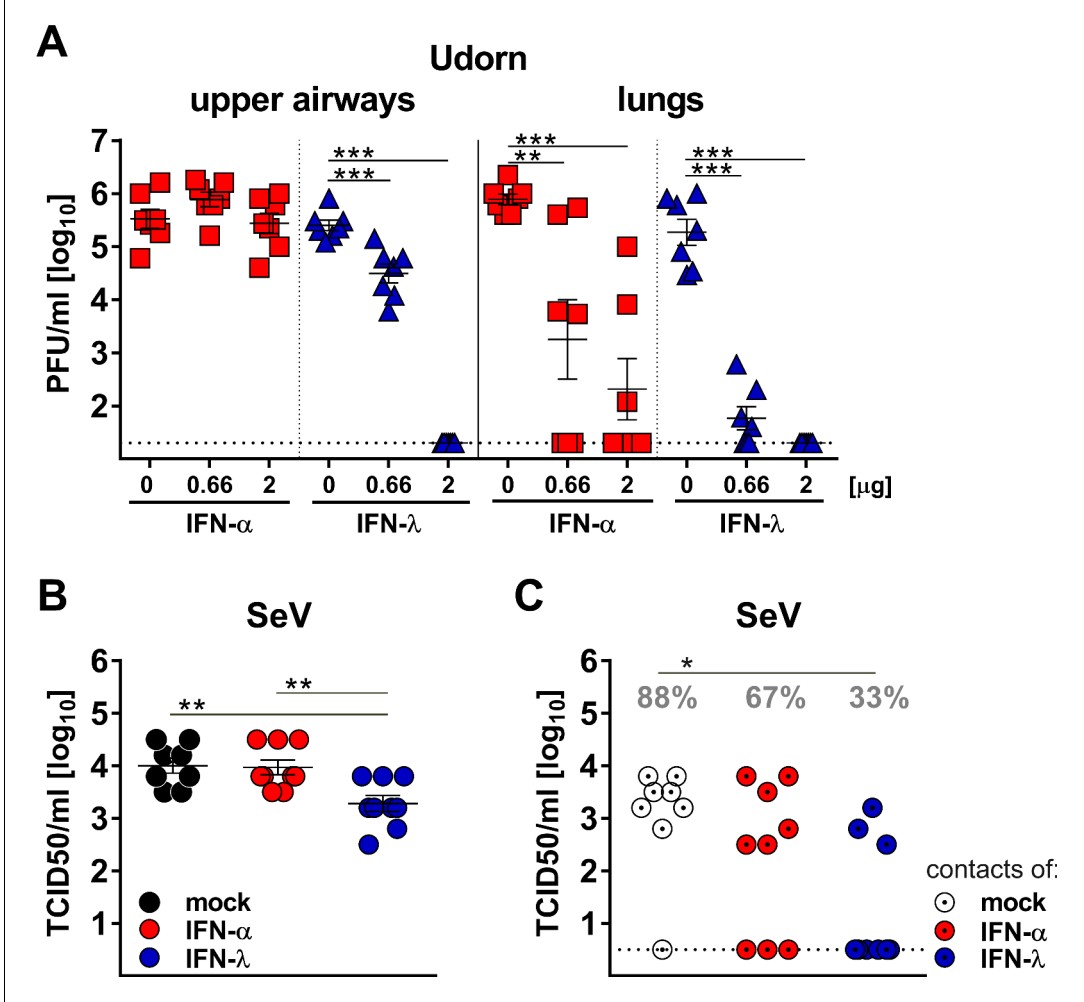

**Figure 5.** Virus defense in the upper respiratory tract strongly relies on IFN-λ. (**A**) Groups (n = 7) of *Ifnar1*[−/−] (blue triangles) and *Ifnlr1*[−/−] mice (red squares) were intranasally treated with 40 µl of saline solution (mock) or with two different doses of IFN-α or IFN-λ as indicated. After 18 hr, the mice were infected with $10^5$ PFU of Udorn in a 40 µl volume. On day three post infection, viral loads in the upper airways and lungs were determined by plaque assay. (**B–C**) Groups of WT mice (n = 8–9) were treated intranasally with 20 µl containing saline (mock), 3 µg of IFN-α or 3 µg of IFN-λ. After 18 hr, the mice were infected intranasally with $10^4$ $TCID_{50}$ of SeV in a volume of 10 µl. At 24 hr post infection, infected mice were cohoused with naive *Ifnar1*[−/−]*Ifnlr1*[−/−] contact mice. On day five post cohousing, all animals were sacrificed and virus titers in the upper airways of directly infected mice (panel B) and contact mice (panel C) were determined by the $TCID_{50}$ method. Grey numbers indicate the calculated frequency of successful virus transmission. Statistical analysis: (**A–B**) One-way ANOVA with Tukey's multiple comparisons; (**C**) Fisher's exact test. Asterisks indicate p-values: ***p<0.001, **p<0.01, *p<0.05. Symbols represent values of individual mice, and bars represent means ± SEM.
DOI: https://doi.org/10.7554/eLife.33354.008

The following figure supplements are available for figure 5:

**Figure supplement 1.** IFN-λ efficiently inhibits influenza virus replication in the upper airways under low dose infection conditions.
DOI: https://doi.org/10.7554/eLife.33354.009

**Figure supplement 2.** IFN-λ and IFN-α have comparable potency on primary airway epithelial cells.
DOI: https://doi.org/10.7554/eLife.33354.010

provide long-lasting antiviral protection in the upper airways, but provided substantial antiviral protection in the lungs (*Figure 6—figure supplement 1*). The short-lived nature of the IFN-α response was also evident from experiments in which *Mx1* gene expression was analyzed in IFN-treated differentiated primary airway epithelial cells derived from mouse tracheae (*Figure 6C*). *Mx1* mRNA levels in IFN-α-treated cells already peaked at 4 hr post onset of treatment and showed a sharp decrease afterwards, nearly reaching baseline levels after 48 hr. In contrast, IFN-λ-mediated induction of *Mx1* gene expression increased until 24 hr post onset of treatment, and even after 72 hr *Mx1* mRNA

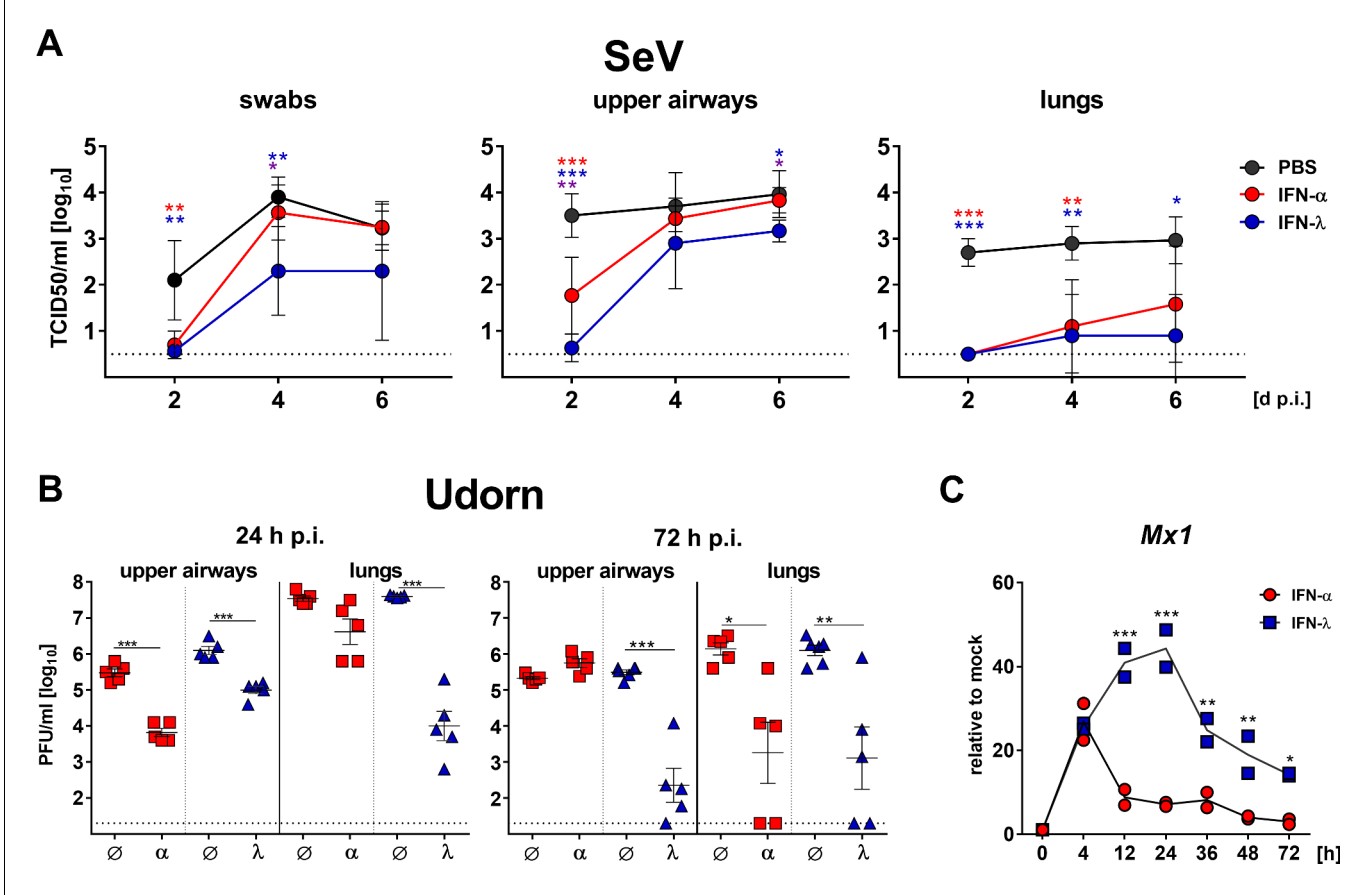

**Figure 6.** IFN-λ but not IFN-α confers long-lasting antiviral protection in the upper airways. (**A**) WT mice (n = 4–5) were treated by the subcutaneous route with 3 µg IFN-α, 3 µg IFN-λ or saline solution 18 hr before intranasal challenge with $10^3$ TCID$_{50}$ of SeV in a 40 µl volume. Groups of mice were sacrificed on days 2, 4 or 6 post infection (dp.i.), and viral titers in nasal swabs (left panel), upper airways (middle panel) and lungs (right panel) were determined by the TCID$_{50}$ method. Symbols represent means ±SD. Red or blue asterisks indicate statistically significant differences between mock and IFN-α- or IFN-λ-treated groups, respectively; purple asterisks indicate differences between IFN-α- or IFN-λ-treated groups. Statistical analysis: One-way ANOVA with Tukey's multiple comparisons; p-values: ***p<0.001, **p<0.01, *p<0.05. (**B**) Ifnar1$^{-/-}$ (blue triangles, n = 5–6) or Ifnlr1$^{-/-}$ mice (red squares, n = 5) were treated by the intranasal route with 2 µg IFN-λ or 2 µg IFN-α, respectively, before intranasal challenge with 4 × $10^5$ PFU of Udorn in a 40 µl volume. Mice treated with saline (∅) served as controls. Mice were sacrificed at 24 hr (left panel) or 72 hr (right panel) post infection (p.i.), and viral titers in the upper airways and lungs were determined by plaque assay. Symbols represent individual mice, and bars represent means ± SEM. Statistical analysis: One-way ANOVA with Tukey's multiple comparisons; p-values: ***p<0.001, **p<0.01, *p<0.05. (**C**) IFN-mediated induction of Mx1 was determined by stimulating differentiated primary airway epithelial cells derived from mouse tracheae for the indicated time points with 1 ng/ml of either IFN-α or IFN-λ. Mx1 induction was assessed by RT-qPCR; values are represented as gene expression levels relative to unstimulated controls (mock). Symbols represent single wells; line indicates mean. Representative data of two independent experiments is shown. Statistical analysis: Two-way ANOVA; asterisks indicate significant differences between IFN-α- and IFN-λ-treated cells. P-values: ***p<0.001, **p<0.01, *p<0.05.

DOI: https://doi.org/10.7554/eLife.33354.011

The following figure supplement is available for figure 6:

**Figure supplement 1.** IFN-λ but not IFN-α confers antiviral protection in the upper airways.

DOI: https://doi.org/10.7554/eLife.33354.012

levels were still close to peak values in IFN-α-treated cells (*Figure 6C*). In summary, these data demonstrated that the antiviral effect of IFN-αB/D especially in the upper airways is surprisingly short-lived, irrespective of the application route.

## A small fraction of epithelial cells in the upper airways of IFN-α-treated mice remains virus-susceptible

Next, we set out to visualize virus-infected cells in the upper airways of mice. At 24 hr post infection with the Udorn virus, many epithelial cells lining the nasal cavity (rostral naso- and maxilloturbinates)

of IFN receptor-deficient mice were strongly positive for viral antigen (*Figure 7A and B*, top panels; *Figure 7—figure supplement 1*). Epithelial cells in other parts of the upper airways showed no sign of viral infection. In *Ifnar1$^{-/-}$* mice that were treated intranasally with 2 µg IFN-λ before infection, no virus-infected cells could be detected by this method (*Figure 7A*). In contrast, in *Ifnlr1$^{-/-}$* mice pre-treated with 2 µg IFN-α, a small number of virus antigen-positive cells were present in most sections of the nasal cavity which included parts of the rostral naso- and maxilloturbinates (*Figure 7B*). These rare virus antigen-positive cells in IFN-α-treated *Ifnlr1$^{-/-}$* mice are most likely epithelial cells, due to their location and expression of the epithelial marker EpCAM. These cells did not contain detectable levels of IFN-inducible nuclear MX1 protein, whereas uninfected neighboring cells did (*Figure 7C*). We concluded from these observations that the upper airways of mice contain some influenza virus-susceptible epithelial cells which appear to rely entirely on IFN-λ for long-lasting antiviral defense.

## Discussion

IFN-λ has previously been recognized as an important component of the innate immune system that limits the replication of viruses infecting epithelial cells of the intestinal tract (*Lazear et al., 2015*; *Wack et al., 2015*). Since *Ifnlr1$^{-/-}$* mice showed only minimally enhanced susceptibility towards influenza viruses compared with wild-type mice in previous studies (*Crotta et al., 2013*; *Galani et al., 2017*; *Mordstein et al., 2008*), it was assumed that the contribution of IFN-λ to protection against respiratory viruses is minor and becomes detectable only when the type I IFN system is defective (*Crotta et al., 2013*; *Mordstein et al., 2008*; *Mordstein et al., 2010a*; *Mordstein et al., 2010b*) or when infections are performed with very low doses of virus (*Galani et al., 2017*). Prior studies neglected the upper respiratory tract, where the decisive battle between invading viruses and the host defense system usually takes place. In most previous studies the inoculum was applied to the airways of the animals in a relatively large volume, which delivers the virus directly to the lungs. In this study we used an experimental setup which ensures that the virus is delivered specifically to the upper airways, thus mimicking the natural infection scenario of respiratory viruses. Our work indicates that previous studies have grossly underestimated the non-redundant antiviral potential of IFN-λ in the respiratory tract, and we demonstrate that IFN-λ is of central importance for antiviral defense of the upper airway mucosa during respiratory virus challenge. Galani et al. similarly concluded from a recent study that IFN-λ mediates non-redundant frontline protection against influenza virus infections (*Galani et al., 2017*). Their work focused exclusively on the IFN-λ-mediated control of viral replication in the lungs. Although our data is compatible with the published findings, it further demonstrates that the non-redundant role of IFN-λ against respiratory viruses is far more prominent in the upper airways than in the lungs. Our results also show that IFN-λ plays a more important role than type I IFN in limiting the transmission of respiratory viruses to naïve contacts and that IFN-λ confers effective and long-lasting protection.

Our new data reveals a compartmentalization of the IFN-based antiviral defense system in the upper respiratory tract that is reminiscent to what was previously described for the intestinal tract (*Mahlakõiv et al., 2015*). Thus, a picture emerges which indicates that IFN-λ is essential for barrier integrity in tissues where viruses most frequently attack the host, and that the type I IFN system ramps up at these sites only after the epithelial barrier has failed to contain the virus. A likely explanation for the evolution of such functional compartmentalization is that this strategy allows shielding the body against minor viral attacks without causing a strong activation of immune cells (*Davidson et al., 2015*; *Davidson et al., 2016*; *Trinchieri, 2010*; *Wack et al., 2015*). Since IFN-λ predominantly acts on epithelial cells rather than immune cells (*Sommereyns et al., 2008*), it is well suited to limit viral replication at epithelial surfaces without inducing a strong inflammatory response commonly associated with type I IFN (*Davidson et al., 2014*; *Davidson et al., 2016*; *Galani et al., 2017*; *Lee-Kirsch, 2017*).

Different mechanisms seem to favor IFN-λ over type I IFN in the antiviral defense of the gut and the respiratory tract. Epithelial cells of the intestinal tract of adult mice express no or only very low levels of functional type I IFN receptors (*Lin et al., 2016*; *Mahlakõiv et al., 2015*) and, consequently, these cells are largely blind to the protective activity of IFN-α and IFN-β. In contrast, replication of influenza and Sendai viruses was clearly inhibited in the upper airways of IFN-α-treated mice during the first days of infection (*Figure 6*), excluding the possibility that airway epithelial cells in general are devoid of functional type I IFN receptors. However, the IFN-α-mediated antiviral protection of

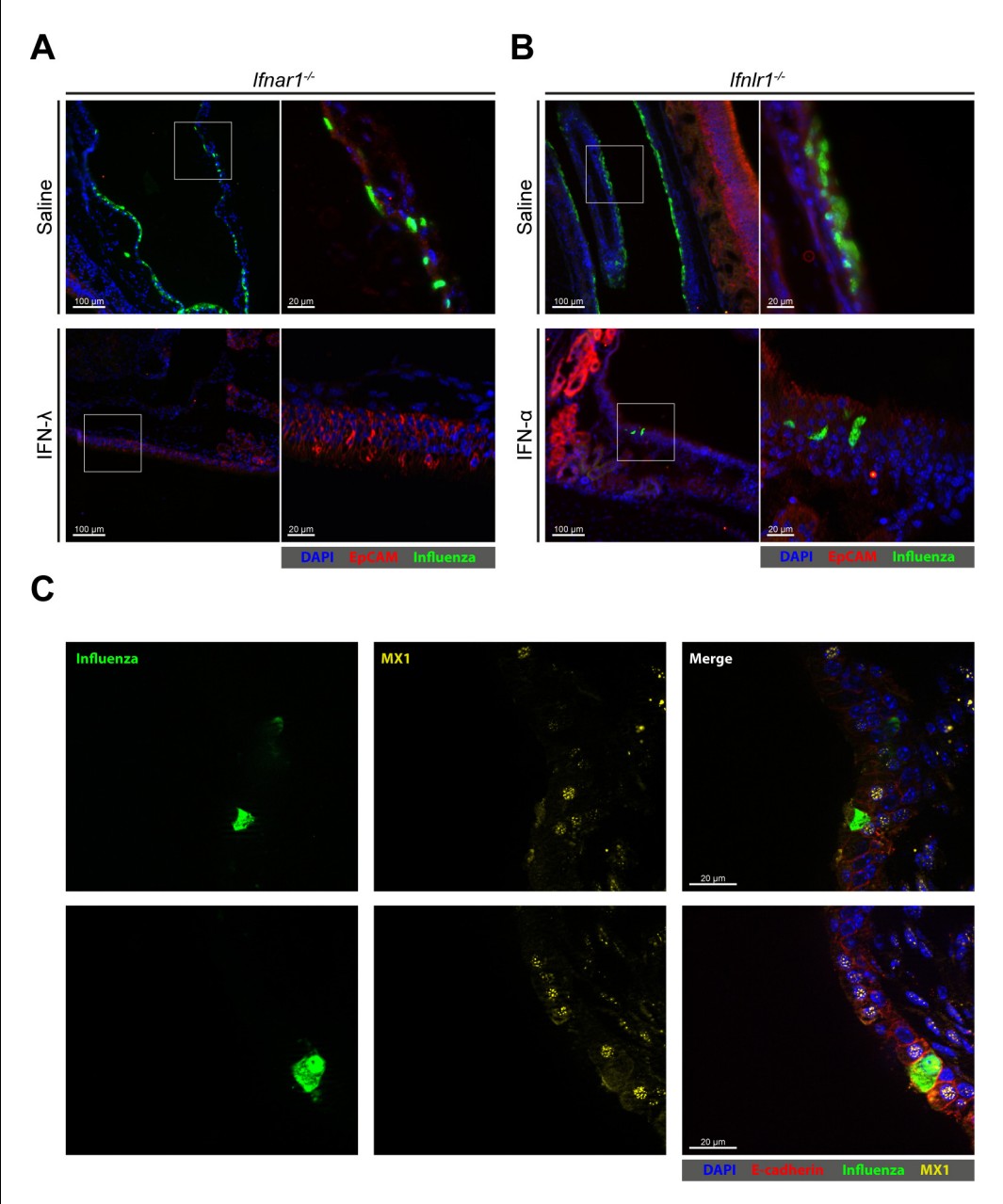

**Figure 7.** Few epithelial cells in upper airways of IFN-α-treated mice remain susceptible to viral infection. (A) *Ifnar1*$^{-/-}$ and (B) *Ifnlr1*$^{-/-}$ mice were treated intranasally with either saline, 2 μg IFN-λ or 2 μg IFN-α before infection with 10$^5$ PFU of Udorn. The animals were sacrificed at 24 hr post infection, and heads were processed for cryosections. Thin-sections were stained for EpCAM (red), influenza virus antigens (green) and DAPI (blue). Merged pictures are presented at low magnification (left panels), and boxed areas are shown at higher magnification (right panels). (C) WT mice were treated intranasally with 2 μg IFN-α before infection with 10$^5$ PFU of Udorn. The animals were sacrificed at 24 hr post infection, and heads were processed for cryosections. Thin-sections were stained for E-cadherin (red), influenza virus antigens (green), MX1 (yellow) and DAPI (blue). Single staining for virus antigen (left panels) and MX1 (middle panels) as well as merged pictures of the same fields (right panels) are shown.
DOI: https://doi.org/10.7554/eLife.33354.013

The following figure supplement is available for figure 7:

**Figure supplement 1.** Udorn replicates preferentially in the rostral naso- and maxilla-turbinates.
DOI: https://doi.org/10.7554/eLife.33354.014

the upper airways, but not the lungs, was surprisingly short-lived and IFN-α-treated and untreated animals contained comparable viral titers in the upper airways at later times post infection. Furthermore, treatment of primary AECs with IFN-α resulted in a very short-lived upregulation of ISG expression. The situation was strikingly different when IFN-λ was applied. IFN-λ treatment resulted in long-lasting ISG induction in primary AECs and sustained antiviral protection of both the lower and the upper respiratory tract in vivo. Analysis at the single cell level revealed that a small number of epithelial cells in the nasal cavity of mice remained virus-susceptible even if the animals were treated with IFN-α. Interestingly, these cells contained no detectable levels of the IFN-inducible MX1 protein, although the animals were treated with a high dose of IFN-α before virus infection (*Figure 7*). Since the antiviral effect of IFN-α in the upper airways was poor irrespective of whether it was applied intranasally or systemically, it is unlikely that IFN-α simply failed to penetrate the mucosa of the upper airways. It remains possible that the influenza virus-susceptible cells in the upper airways of IFN-α-treated mice represent a rare specialized epithelial cell type which possesses no functional type I IFN receptors (*Lin et al., 2016*; *Mahlakõiv et al., 2015*). More likely, however, these cells contain high levels of factors that inhibit type I but not type III IFN receptor signaling such as USP18, SOCS1 and SOCS3 (*François-Newton et al., 2011*; *Makowska et al., 2011*; *Porritt and Hertzog, 2015*). IFNAR1 can be rapidly internalized and degraded upon activation (*Fuchs, 2013*), while no such effects were reported for the IFN-λ receptor complex. Thus, it remains possible that negative feed-back regulation of IFNAR1 expression is exceptionally strong in epithelial cells of the upper airways. It is important to note that unhindered virus replication in these rare IFN-α-unresponsive cells might initiate a second wave of virus infection in the upper airways shortly after the antiviral state induced by the IFN-α treatment has waned, which could provide an explanation for the transient antiviral effect of IFN-α in the upper airways.

Since IFN-λ exhibits selective, sustained and highly powerful antiviral activity in the upper airways that efficiently reduces morbidity and mortality (*Davidson et al., 2014*) as well as transmission of respiratory viruses in mice (our current work), we suggest to consider IFN-λ not only for treating diseased patients (*Davidson et al., 2014*; *Davidson et al., 2015*; *Davidson et al., 2016*; *Wack et al., 2015*) but also for the management of respiratory virus outbreaks in community settings. According to our results, such clinical use of IFN-λ may not only improve the health status of patients with respiratory symptoms, but is also expected to strongly inhibit virus transmission from infected individuals to healthy contacts. Clinical studies indicate that the side effects of IFN-λ are minimal (*Muir et al., 2014*). Thus, the use of IFN-λ as prophylactic drug seems justified in health care units during epidemics with respiratory viruses as an effort to protect people who are not fully protected by vaccines.

## Materials and methods

### Viruses

The following influenza A virus strains were used: A/Seal/Massachusetts/1/1980 (H7N7) designated SC35M, A/Hong Kong/8/68 (H3N2) designated HK68, and A/Udorn/72 (H3N2) designated Udorn. SC35M represents a mouse-adapted variant of an avian-like H7N7 virus that was originally isolated from a diseased seal. HK68 and Udorn are human virus isolates and have no passage history in mice. MDCK cells were used for the preparation of influenza virus stocks and for virus titration by plaque assay. Recombinant Sendai virus (SeV) expressing green fluorescent protein (GFP) was grown in the allantoic cavity of 9-day-old embryonated chicken eggs for 3 days at 33°C (*Strähle et al., 2007*). SeV titers of mouse organ extracts were determined by counting GFP-positive foci after infection of Vero cells at different dilutions. Polio virus type I Mahoney, a strain derived from the infectious cDNA clone pOM, was used in this study (*Shiroki et al., 1995*). Polio virus titration by plaque assay was performed on Vero cells.

### Mice and viral infections

B6.A2G-*Mx1* mice (designated WT) are C57BL/6 mice carrying functional *Mx1* alleles (*Mordstein et al., 2008*). B6.A2G-*Mx1-Ifnar1*$^{-/-}$ mice (designated *Ifnar1*$^{-/-}$) lack functional type I IFN receptors, B6.A2G-*Mx1-Ifnlr1*$^{-/-}$ mice (designated *Ifnlr1*$^{-/-}$) lack functional IFN-λ receptors. *Ifnar1*$^{-/-}$ *Ifnlr1*$^{-/-}$ lack both IFN receptor systems (*Mordstein et al., 2008*). All mice used in this

study were bred locally in our facility or purchased from Janvier (Strasbourg). Animals were handled in accordance with guidelines of the Federation for Laboratory Animal Science Associations and the national animal welfare body. Animal experiments were performed in compliance with the German animal protection laws and were approved by the university's animal welfare committee (Regierungspräsidium Freiburg; permit G-15/59). Mice used for the experiments were 6 to 13 weeks old.

To achieve infection of the entire respiratory tract, virus in OptiMEM medium containing 0.3% BSA was administered intranasally to ketamine/xylazine-anesthetized animals in a volume of 40 µl. For selective infection of the upper respiratory tract, the virus was administered intranasally under light anesthesia (3% isoflurane in oxygen) in a volume of 10 µl. For transmission experiments, infected mice of different genetic backgrounds were cohoused with naïve $Ifnar1^{-/-} Ifnlr1^{-/-}$ contact mice in a fresh cage for 3–5 days starting at 24 hr post infection. Transmission experiments with Udorn and HK68 were performed by cohousing three index mice with four naïve contact mice, whereas transmission experiments with SeV were performed by cohousing individual index mice with one contact mouse each.

To determine viral titers in various organs, mice were sacrificed by cervical dislocation at the indicated time points. The upper airways, tracheae and lungs were harvested and stored at −80°C until further proceedings. For immune staining, complete heads without skin and fur were collected and fixed by incubation in 4% formaldehyde at 4°C for 16 hr. To determine viral titers, organs were homogenized in 800 µl ice-cold PBS using FastPrep tubes, spheres, and homogenizer (MP Biomedicals, USA). Two cycles of homogenization at 6.5 m/s for 16 s were performed, with samples resting on ice in between. Homogenates were centrifuged at 2300 x g for 5 min at 4°C, supernatants were collected and 10-fold serial dilutions in Opti-MEM with 0.3% BSA were applied to MDCK or Vero cells for plaque assay or determination of $TCID_{50}$, respectively.

## IFN treatment of mice

Mice were treated with the indicated doses of either human IFN-αB/D which is active in mice (*Horisberger and de Staritzky, 1987*) or recombinant mouse IFN-λ2 (*Dellgren et al., 2009*) by either the intranasal (20 or 40 µl) or the subcutaneous (100 µl) route.

## Immunohistochemistry

Formaldehyde-fixed heads were decalcified in Osteosoft at room temperature on a roller shaker for 96 hr. Samples were then incubated in 15% sucrose at 4°C for 4 hr, followed by incubation in 30% sucrose overnight. Samples were embedded with Tissue-Tek O.C.T. compound and stored at −80°C until use. 4 µm sections were prepared using a cryotome, dried overnight at 37°C and permeabilized for 5 min with 0.5% Triton-X in PBS. Blocking was performed with 10% donkey normal serum in PBS for 45 min. Primary antibodies were goat anti-influenza A (Purified, AbD Serotec (OBT1551)), rabbit anti-MX1 (anti-AP5 peptide [*Meier et al., 1988*]), rat anti-EpCAM (Purified, BD Biosciences (552370)) and mouse anti-E-Cadherin A647 (Purified, BD Biosciences (560062)). Secondary antibodies were donkey anti-goat A488 (Purified, Jackson Immuno Research (705-545-147)), donkey anti-rat A647 (Purified, Abcam (ab150155)) and donkey anti-rabbit A555 (Purified, Invitrogen (A-31572)). Nuclei were stained with DAPI and slides were mounted with FluorSave reagent.

## Primary mouse tracheal epithelial cell culture

Isolation and culturing of primary mouse airway epithelial cells were performed as previously described (*Crotta et al., 2013*). Briefly, cells were isolated from the tracheae of WT mice by enzymatic treatment and seeded onto a 0.4 µm pore size clear polyester membrane (Corning) coated with a collagen solution. At confluence, the medium was removed from the upper chamber to establish an air-liquid interface (ALI). Fully differentiated, 7–10 day-old post-ALI cultures were routinely used for experiments. For analysis of the biological activity of IFN-αB/D and IFN-λ2, cells were stimulated for 4 hr by basolateral supplementation of the indicated IFN concentrations. For continuous treatment, cells were stimulated by basolateral supplementation of 1 ng/ml IFN-α B/D or IFN-λ2. Media containing IFN were refreshed every 24 hr.

## RNA-isolation and RT-qPCR

AEC cultures were lysed directly in the transwells using the RNeasy Plus Mini kit (Qiagen) according to the manufacturer's instructions. Snouts were homogenized in 800 µl peqGOLF TriFast$^{TM}$ FL using FastPrep tubes, spheres, and homogenizer (MP Biomedicals, USA). Four cycles of homogenization at 6.5 m/s for 16 s were performed and samples subsequently centrifuged at 12,000 x g for 10 min at 4°C. Supernatants were diluted (1:2.5) in peqGOLF TriFast FL, combined with 200 µl BCP (1-Bromo-3-chlorpropane, Sigma) per ml TriFast$^{TM}$ FL and phase separation performed with 12,000 x g for 15 min at 4°C. Supernatants were combined with one volume ethanol (Sigma) and RNA was isolated using the RNeasy Plus Mini kit according to the manufacturer's instructions starting with the RNeasy spin column. DNA was removed by using DNAse I, Amplification Grade (Invitrogen). cDNA was generated using the Thermoscript RT-PCR system, following the manufacturer's instructions (Invitrogen). The cDNA served as a template for the amplification of genes of interest (IAV-M1: forward: 5'-AAGACCAATCCTGTCACCTCTGA-3'; reverse: 5'-CAAAGCGTCTACGCTGCAGTCC-3', *Ifnl2/3* (mm0420156_gH, Applied Biosystems), *Ifnb1*: forward: 5'-CCTGGAGCAGCTGAATGGAA-3'; reverse: 5'-CACTGTCTGCTGGTGGAGTTCATC-3'; probe: 5'-[6FAM]CCTACAGGGCGGACTTCAAG [BHQ1]—3', *Ifna4* (QT01774353, QuantiTect Primer Assay, Qiagen), *Isg15*: forward: 5'-GAGC TAGAGCCTGCAGCAAT-3'; reverse: 5'-TTCTGGGCAATCTGCTTCTT-3', *Stat1*: forward: 5'-TCACAG TGGTTCGAGCTTCAG-3'; reverse: 5'-CGAGACATCATAGGCAGCGTG-3', *Mx1*: forward: 5'-TC TGAGGAGAGCCAGACGAT-3'; reverse: 5'-ACTCTGGTCCCCAATGACAG-3' and *Hprt* (mm00446968_m1, Applied Biosystems)) by real-time PCR, using TaqMan Gene Expression Assays (Applied Biosystems), Universal PCR Master Mix (Applied Biosystems) and the ABI-Prism 7900 sequence detection system (Applied Biosystems). The increase in mRNA expression was determined by the $2^{-\Delta Ct}$ method relative to the expression of *Hprt* or by the $2^{-\Delta\Delta Ct}$ method relative to mock.

## Study design and statistical analyses

Mice were randomly assigned to the various experimental groups, but the study was not blinded. Group sizes and endpoints were predefined based on results from suitable pilot experiments. Comparison between more than two groups was evaluated using one-way analysis of variance (ANOVA) with Tukey's multiple comparisons. Two-way ANOVA with Tukey's multiple comparison was used to evaluate more than two groups at different time points. All tests were performed using $\log_{10}$ transformed values for viral titers or relative gene expression. Transmission and viral spread frequencies were analyzed by Fisher's exact test. P-values considered to indicate a significant difference are indicated in the figures as follows: *p<0.05; **p<0.01; ***p<0.001. GraphPad Prism six software (Graph-Pad Software, USA) was used for statistical analysis.

## Acknowledgements

We thank Annette Ohnemus for technical assistance, Otto Haller and Andreas Wack for excellent comments on the manuscript, and Kerry Mills for proof reading. Funding This work was supported by the Deutsche Forschungsgemeinschaft (Grant STA 338/15–1 to PS), the European commission (UniVacFlu to PS), the Danish Council for Independent Research, Medical Research (Grant 11-107588 to RH), the Novo Nordisk Foundation (Grant NNF15OC0017902 to RH), and the Swiss National Fund (31003A_163129 to DG).

## Additional information

### Funding

| Funder | Grant reference number | Author |
|---|---|---|
| Novo Nordisk | NNF15OC0017902 | Rune Hartmann |
| Danish Council for Independent Research-Medical Research | 11-107588 | Rune Hartmann |
| Schweizerischer Nationalfonds zur Förderung der Wissenschaftlichen Forschung | 31003A_163129 | Dominique Garcin |

| Deutsche Forschungsge-meinschaft | STA 338/15–1 | Peter Staeheli |
| European Commission | UniVacFlu | Peter Staeheli |

The funders had no role in study design, data collection and interpretation, or the decision to submit the work for publication.

## Author contributions

Jonas Klinkhammer, Data curation, Formal analysis, Investigation, Visualization, Methodology; Daniel Schnepf, Data curation, Formal analysis, Supervision, Validation, Investigation, Visualization, Methodology, Writing—review and editing; Liang Ye, Data curation, Formal analysis, Investigation, Methodology; Marilena Schwaderlapp, Data curation; Hans Henrik Gad, Rune Hartmann, Dominique Garcin, Resources, Methodology; Tanel Mahlakõiv, Data curation, Formal analysis, Supervision; Peter Staeheli, Conceptualization, Supervision, Funding acquisition, Investigation, Writing—original draft, Project administration, Writing—review and editing

## Author ORCIDs

Hans Henrik Gad http://orcid.org/0000-0001-8449-1115
Dominique Garcin http://orcid.org/0000-0003-1556-897X
Peter Staeheli http://orcid.org/0000-0001-7057-6177

## Ethics

Animal experimentation: Animals were handled in accordance with guidelines of the Federation for Laboratory Animal Science Associations and the national animal welfare body. Animal experiments were performed in compliance with the German animal protection laws and were approved by the university's animal welfare committee (Regierungspräsidium Freiburg; permit G-15/59).

## Decision letter and Author response

Decision letter https://doi.org/10.7554/eLife.33354.017
Author response https://doi.org/10.7554/eLife.33354.018

## Additional files

### Supplementary files

• Transparent reporting form
DOI: https://doi.org/10.7554/eLife.33354.015

### Data availability

All data generated or analyzed during this study are included in the manuscript and supporting files.

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
