## [Decision Letter]

Thank you for submitting your article "IFN-λ prevents influenza virus spread from the upper airways to the lungs and limits virus transmission" for consideration by *eLife*. Your article has been favorably reviewed by three peer reviewers, and the evaluation has been overseen by a Reviewing Editor and Wenhui Li as the Senior Editor. The reviewers have opted to remain anonymous.

The reviewers have discussed the reviews with one another and the Reviewing Editor has drafted this decision to help you prepare a revised submission.

Summary:

This study shows that IFN-λ has non-redundant antiviral effects in the upper respiratory tract, leading to a unique role in controlling virus transmission. The work presented has great merit as virus transmission is a fundamental, yet understudied, parameter in epidemic spread of influenza virus. The importance of IFN-λ in the upper airways for the control of influenza is convincingly shown using different influenza strains. The existence of cells in the URT that are refractory to type I IFN but presumably responsive to IFN-λ is of great relevance.

The use of gene-targeted mice combined with a meaningful model of virus transmission to identify IFN-λ as a crucial protective factor; the use of small-volume inoculum to mimic better infection in humans; the reconciliation of previous, apparently contradictory data obtained by higher volume infection; and a conclusive demonstration of a clear-cut non-redundant role for IFN-λ in the lung are the strengths of this study.

Moreover, the information about the timing of the different IFN responses in vivo and the effect of dosage and anatomy on the balance between virus and host response will be of great interest to experimentalists and modelers alike.

The manuscript is well organized, clearly written and the overall experimental design and data analysis are appropriate. The conclusions are mostly supported by the presented data. A limitation is that only measurements of virus titers are performed in the context of infection while information on IFN production or ISG induction is missing.

Essential revisions:

1) Provide some measurements on how much type I or type III IFN is being produced in the airways. There are no measurements shown on how much type I or type III IFN is being produced in the airways. This data is critical to understand the differences between IFN-α/β and IFN-λ systems during infection.

2) Quantify ISG induction in the mouse airway. The authors have not quantified ISG induction in the mouse airway. It would be informative to measure Mx mRNA in the mouse lung samples.

There do seem to be individual cells in the airway (Figure 6C) that are responding to infection by expressing the Mx protein. Do these cells express only IFN-λ receptors? Some further combinations of infection, IFN treatment and staining for receptors and response genes as shown in Figure 6 would be extremely informative. For example, are the cells stained for virus in Figure 6C also seen in IFN-λ R-/- mice?

3) Provide some data on disease outcome. One of the main differences between the IFN type I and type III systems is the cellular/tissue distribution of the receptor, which limits the range of response for type III IFN and the systemic inflammation that usually comes with type I IFN. Did the authors observe any differences in immunopathology between infected *ifnlr^-/-^* and *ifnar^-/-^* infected mice? Did the author measure levels of pro-inflammatory cytokines in those mice? Some information on morbidity or mortality would help the interpretation.

4) Expand the introduction to clearly highlight the advances of the current study relative to the article by Galini and colleagues.

5) Discuss the potential mechanistic differences between IFN-α and IFN-λ.

– The comparisons with IFN-α are sometime overlooked. For example, Figure 2A shows no difference in virus shedding from WTR or IFNAR-/- animals and yet the transmission data showed only 17% transmission from WT but 42% from IFNAR-/-. Please comment.

– In subsection “IFN-λ confers long-lasting virus protection in the upper airways, whereas IFN-α-mediated protection is short-lived”, the short-lived response to IFN-α is mentioned. Might this be explained by a dosage difference? The peak virus titer in SeV infected lungs is less than that in URT and swabs. IFN-α could just be better at control the lower dose of virus at the lung.

– The authors take care to control the amount of IFNs applied by titrating them on primary mouse epithelial cells. At 4 hours post infection, the two IFNs types look equivalent but at later time points they do differ (Figure 5C). What does that mean for the in vivo results?

6) Expand the discussion to include the mechanisms responsible for different responses especially short lived IFN-α response.

---

## [Author Response]

Essential revisions:1) Provide some measurements on how much type I or type III IFN is being produced in the airways. There are no measurements shown on how much type I or type III IFN is being produced in the airways. This data is critical to understand the differences between IFN-α/β and IFN-λ systems during infection.

We agree with the reviewers that more information on the synthesis of type I and type III IFN in the upper airways of virus-infected mice might help to explain our findings. Accordingly, we employed commercial ELISA kits to measure IFN-λ in lysates prepared from the snout tissue. It turned out that these ELISA kits are not suitable for a proper analysis of IFN levels in tissue homogenates due to strong non-specific background signals. It should be noted that these assays worked well for serum samples and bronchoalveolar lavage fluids. Unfortunately, however, measuring IFN levels in the latter samples does not solve the open questions of our project.

The only method that worked for us was RT-qPCR. Data presented in a new Figure 2A shows that baseline expression of IFN-λ genes is significantly lower in the upper airways of *Ifnar1^-/-^* mice compared with wild-type and *Ifnlr1^-/-^* mice. As a result of reduced baseline expression of the IFN-λ gene, *Ifnar1^-/-^* mice also exhibit low baseline expression of the IFN-stimulated *Mx1* gene which may explain why *Ifnar1^-/-^* mice show enhanced susceptibility to influenza viruses even though they possess a fully functional IFN-λ system. These new findings are described in subsection “IFN-λ prevents influenza virus spread from the upper respiratory tract to the lungs” of our revised manuscript.

2) Quantify ISG induction in the mouse airway. The authors have not quantified ISG induction in the mouse airway. It would be informative to measure Mx mRNA in the mouse lung samples.

In the new Figure 2B we show induction of IFN genes and a representative ISG (*Mx1*) in the upper airways of infected mice. Baseline expression of IFN-λ mRNA was remarkably high in WT mice, and expression was only moderately induced after virus infection. *Ifnar1*^-/-^ mice had significantly lower levels of IFN-λ mRNA before infection. Due to technical issues (as explained above) high baseline expression of IFN-λ in snouts of WT mice could not be confirmed on the protein level. Interestingly, expression of the IFN-λ genes in *Ifnar1^-/-^* mice reached comparable levels as in WT or *Ifnlr1^-^*^/-^ mice within two days post virus infection. These data indicate that functional type I IFN signaling is needed to prime cells for proper baseline expression of the IFN-λ genes which contributes to early virus defense in the upper airways. These important findings are described in subsection “IFN-λ prevents influenza virus spread from the upper respiratory tract to the lungs” of the revised manuscript.

As requested by the reviewer, we also measured *Mx1* expression in lung samples of our mice. We found no significant induction of *Mx1* expression in lung tissue of WT mice within the first three days after selective upper airway infection. These results indicate that IFN-λ is acting locally and stops virus spread from the upper airways to the lungs without triggering a systemic IFN response. We decided against including these negative data in our revised manuscript, mainly because the focus of our work is on the upper airways.

There do seem to be individual cells in the airway (Figure 6C) that are responding to infection by expressing the Mx protein. Do these cells express only IFN-λ receptors? Some further combinations of infection, IFN treatment and staining for receptors and response genes as shown in Figure 6 would be extremely informative. For example, are the cells stained for virus in Figure 6C also seen in IFN-λ R-/- mice?

Figure 6C (Figure 7C of the revised manuscript) shows snout tissue of a wild-type mouse that was treated with IFN-α before infection with Udorn virus. We concluded from this figure that a small number of epithelial cells in the upper airways do not respond to IFN-α and that these cells seem to remain susceptible to virus infection in spite of an IFN-α pretreatment. We would like to emphasize that similar experiments with *Ifnar1^-/-^* and *Ifnlr1^-/-^* mice that were treated before virus infection with IFN-λ and IFN-α, respectively, are shown in Figure 7A and 7B (formerly 6A and 6B). They clearly illustrate that IFN-λ blocked replication of Udorn virus very efficiently, whereas IFN-α did not suppress virus replication in all epithelial cells of the upper airways neither in WT nor in *Ifnlr1*^-/-^.

3) Provide some data on disease outcome. One of the main differences between the IFN type I and type III systems is the cellular/tissue distribution of the receptor, which limits the range of response for type III IFN and the systemic inflammation that usually comes with type I IFN. Did the authors observe any differences in immunopathology between infected ifnlr^-/-^ and ifnar^-/-^ infected mice? Did the author measure levels of pro-inflammatory cytokines in those mice? Some information on morbidity or mortality would help the interpretation.

Apparently, we did not stress sufficiently well that our mouse-based virus transmission model is based on H3N2 influenza virus strains that do not cause overt disease in mice. Further, most of our infection experiments were designed to deliver the virus inoculum selectively to the upper airways rather than to the lungs. It is well known (Ivinson et al., 2017) that such targeted virus infections are not causing disease in mice even if virus strains are used which are highly pathogenic when delivered directly to the lungs. Most likely for these technical reasons we did not observe any differences in immunopathology between *Ifnar1^-/-^* and *Ifnlr1^-/-^* mice. We modified the text in subsection “IFN-λ prevents influenza virus spread from the upper respiratory tract to the lungs” of the revised manuscript to better explain our model.

4) Expand the introduction to clearly highlight the advances of the current study relative to the article by Galini and colleagues.

We agree that, at first glance, our work shows similarity to a recently published paper by Galani and coworkers. However, there are important differences between the design and the conclusions of the two studies.

First, we used a protocol in which virus infection selectively starts in the upper airways, mimicking natural virus contraction. Galani et al., in contrast, used a low virus dose to inoculate the whole respiratory tract, as it is conventionally done in respiratory virus studies.

Second, we show that the IFN-λ response stops the virus spread already in the respiratory epithelia of the upper airways, preventing infection of the lungs altogether. We demonstrate that the non-redundant role of IFN-λ is most pronounced in the upper airways where the decisive battle between invading respiratory viruses and the host immune defense takes place under natural conditions. We demonstrate that increased amounts of infectious particles are secreted if the IFN-λ system is defective. This enhances the likelihood that the virus will successfully spread to the lungs of the infected individual. In contrast, Galani et al., exclusively focused on the control of viral replication after it has reached the lungs, and on the immune pathology in the infected lungs.

Third, we demonstrate that IFN-λ-mediated control of viral replication in the upper airways is critical for limiting virus transmission. High amounts of infectious particles are secreted from the upper airways which enhance the probability that the virus is successfully transmitted to a new host. Galani et al., did not assess the role of IFN-λ for virus shedding and contact transmission.

Forth, we used mice with functional *Mx1* alleles for our studies. In such mice, IFN-mediated antiviral effects against influenza viruses are much more pronounced than in mice lacking functional *Mx1* alleles, providing a broader experimental window which permits detecting small differences in antiviral activity of different IFN subtypes with higher sensitivity. Galani et al., used mice with defective *Mx1* alleles. As a consequence, the protective effects of IFN-λ were more pronounced and more sustained in our study compared with the study published by Galani and coworkers.

As requested by the reviewers, we extended the Introduction and the Discussion section of our manuscript to more clearly highlight the advances of our new work relative to the earlier study published by Galani and coworkers.

5) Discuss the potential mechanistic differences between IFN-α and IFN-λ.– The comparisons with IFN-α are sometime overlooked. For example, Figure 2A shows no difference in virus shedding from WTR or IFNAR-/- animals and yet the transmission data showed only 17% transmission from WT but 42% from IFNAR^-/-^. Please comment.

The main reason why we do not put emphasis on the type I IFN data shown in Figure 3A (formerly 2A) is that the protective effect of IFN-α was only very mild. As correctly noted by the reviewer, virus shedding was not enhanced in *Ifnar1^-/-^* mice. Furthermore, although the rate of Udorn virus transmission was slightly enhanced in the experiment shown in figure 4A, this difference did not reach statistical significance and was not observed in transmission experiments with the HK68 virus strain (Figure 4B). The same holds true for SeV: neither virus shedding (Figure 3B) nor transmission (Figure 4C) was significantly enhanced in *Ifnar1^-/-^* mice compared with wild-type animals.

– In subsection “IFN-λ confers long-lasting virus protection in the upper airways, whereas IFN-α-mediated protection is short-lived”, the short-lived response to IFN α is mentioned. Might this be explained by a dosage difference? The peak virus titer in SeV infected lungs is less than that in URT and swabs. IFN-α could just be better at control the lower dose of virus at the lung.

The short-lived nature of the IFN-α effect cannot be explained by a dosage difference. We used identical amounts of IFN-α and IFN-λ for the treatments, and we showed in Figure 5—figure supplement 2 that both IFN preparations have similar biological activities. It is correct that SeV grew to lower titers in the lungs as compared with the upper airways. However, this observation is not relevant for our argument. In the experiment shown in Figure 6, we infected the whole respiratory tract by applying the virus in a volume of 40 µl. Thus, at the earliest times post infection, virus levels were equally low in all parts of the respiratory tract. Since the IFN treatment of the animals was performed before virus infection, our data shown in Figure 6 demonstrates that IFN-α is less effective in preventing virus growth in the upper airways than in the lungs. The short-lived nature of the IFN-α response is shown in Figure 6B and 6C.

– The authors take care to control the amount of IFNs applied by titrating them on primary mouse epithelial cells. At 4 hours post infection, the two IFNs types look equivalent but at later time points they do differ (Figure 5C). What does that mean for the in vivo results?

The discovery that the biological effect of IFN-α is short-lived in airway epithelial cells is important. It provides a good explanation for why IFN-α is less potent than IFN-λ in controlling virus growth in the upper airways. Figure 6B formally excludes a second alternative explanation of our findings, namely the possibility that upper airway epithelial cells might lack functional receptors for type I IFN. We extended the Discussion section to highlight this important finding.

6) Expand the discussion to include the mechanisms responsible for different responses especially short lived IFN-α response.

We discussed possible mechanisms explaining the short-lived nature of the IFN-α response (see Discussion section).